# Dietary Patterns in Relation to Asthma and Wheeze Among Adolescents in a South African Rural Community

**DOI:** 10.3390/ijerph22040502

**Published:** 2025-03-26

**Authors:** Funzani Rathogwa-Takalani, Thabelo Rodney Mudau, Sean Mark Patrick, Joyce Shirinde, Kuku Voyi

**Affiliations:** 1Department of Advanced Nursing Science, Faculty of Health Sciences, University of Venda, Thohoyandou 0950, South Africa; 2School of Health Systems and Public Health, University of Pretoria, Pretoria 0001, South Africa; sean.patrick@up.ac.za (S.M.P.); joyce.shirinde@up.ac.za (J.S.); kuku.voyi@up.ac.za (K.V.); 3Centre for Environmental and Occupational Health Research, School of Public Health, University of Cape Town, Cape Town 7925, South Africa; thabelo.mudau@uct.ac.za; 4Environmental Chemical Pollution and Health Research Unit, University of Pretoria, Pretoria 0001, South Africa

**Keywords:** asthma, wheeze, dietary patterns, adolescents, South Africa

## Abstract

Background: The rise of asthma prevalence in recent decades has been attributed to changes in dietary patterns, especially in developing countries. Studies have also suggested that dietary patterns play an important role in both asthma development and management. This study aimed to investigate the association between consumption of various foods and environmental factors with asthma and wheeze among adolescents. Methods: A self-administered standardized International Study of Asthma and Allergies in Childhood (ISAAC) questionnaire was used to collect data on demographics, respiratory health, exposure to air pollution, and diet on n = 2855 adolescents residing in Vhembe District, South Africa. Results: The prevalence of asthma and wheeze were 18.91% and 37.69%, respectively. Consuming various foods such as fast foods (OR = 1.41; 95% CI: 1.06–1.88), bread (OR = 0.60; 95% CI: 0.45–1.81), pasta (OR = 1.39; 95% CI: 1.06–1.84), seafood (OR = 1.79; 95% CI: 0.65–1.24), and nuts (OR = 0.85; 95% CI: 0.65–1.12) were significantly associated with asthma in the crude logistic regression analysis. Further analysis in the multiple regression model indicated a strong association of asthma with consumption of nuts (OR = 1.55; 95% CI: 1.11–2.17), seafood (OR = 1.60; 95% CI: 1.03–2.49), and cereal (OR = 0.67; 95% CI: 0.45–0.99). In relation to wheeze, consumption of meat (red) (OR = 0.77; 95% CI: 0.60–0.99) was a protective factor in the crude analysis. The multiple logistic regression model showed that, seafood (OR = 0.76; 95% CI: 0.59–0.96), fruit (OR = 0.55; 95% CI: 0.32–0.94), nuts (OR = 1.88; 95% CI: 1.50–2.66), and olive oil (OR = 1.48; 95% CI: 1.09–2.00) were significantly associated with wheeze. Conclusion: Diet plays a major role in respiratory health, especially in asthma and wheeze. Dietary changes may play a role in reducing the burden of asthma and other respiratory symptoms in adolescents.

## 1. Introduction

Asthma is a chronic respiratory disease that is associated with airway inflammation and affects millions of people worldwide [1,2,3,4,5]. Wheeze is a symptom of asthma, and a common condition characterized by a continuous high-pitched sound that emanates from the chest during exhalation [6,7]. The significant increase in asthma prevalence in recent years has been linked to factors such as smoking, air pollution, and seasonal variations, while others have linked it to the change in diet [8,9]. Various studies are showing an increase in the consumption of an unhealthy diet, in particular fast foods [8,10]. In South Africa, similar to other developing countries, people have adopted a western lifestyle which is mainly characterized by the frequent consumption of unhealthy foods [8].

Although research has focused on avoiding environmental risk factors such as air pollution, dampness and molds, and exposure to smoke to prevent asthma and its exacerbations, strong evidence exists to show that diet plays a critical role in respiratory health [8,11]. While some studies focused on healthy eating habits, others have investigated exposure to diverse microorganisms in the environment as they are said to have influence in asthma and wheeze especially in early childhood [11,12,13].

Despite indoor and outdoor air pollution, inhaled allergens and exposure to smoke are well known triggers of asthma and wheeze [14]. Recent studies have reported that the significant increase in the prevalence of asthma is not only related to environmental factors but also lifestyle changes including diet [2,8]. The Mediterranean diet is considered an adequate diet as it consists of mainly fruits, vegetables, eggs, fish, and healthy fats. This diet is known to have anti-inflammatory properties and could also be a protective factor to mitigate asthma and wheeze [2,14,15]. Contrary to this, a western diet consisting of processed foods, desserts, red meats, and fast foods that have pro-inflammatory effects is a risk factor for asthma and wheeze [2,8,9]. Recent studies recognize several dietary factors as potential contributors to the development and management of asthma [1,9]. Although this aspect is not well studied, research studies do suggest that diet may modulate both inflammation and the immune system which are important factors in the development and severity of asthma symptoms.

Previously conducted studies on diet and its association with wheeze and asthma have focused on specific foods, e.g., a study conducted in Gauteng, South Africa, focused on the frequency of consuming fast foods and the association with wheeze and asthma [8]. This study reported a significant association between fast food consumption and asthma and wheeze, suggesting that an unhealthy diet is a contributor to asthma and wheeze, and, therefore, healthy diets should be encouraged [8].

Moreover, a study conducted by Chatzi et al. [16] showed that consuming fruits and vegetables may reduce the risk of asthma and wheeze. Furthermore, the study highlights that consuming fruits and vegetables for extended periods was associated with better asthma management and a lower risk of allergy symptoms in asthmatic children [2,16]. Another study emphasized that consumption of a diet high in processed foods and total saturated fats may increase inflammation of the airways in asthmatics, making asthma symptoms more severe [17].

Although some research has investigated the association of diet with asthma and wheeze, a study to investigate the association of dietary patterns with asthma and wheeze has not been conducted in the Limpopo province, South Africa. Hence, further studies are needed to better understand the association between dietary patterns, asthma, and wheeze in this setting. The purpose of this study was to investigate the association between consumption of different foods with asthma and wheezing among adolescents.

## 2. Materials and Methods

### 2.1. Study Design and Participants

This study was a cross-sectional study that included 13–14 years old adolescents who reside in Vhembe district, South Africa, and attended school in the selected communities. This student-based study took place from June 2023 to August 2024. Randomly selected schools (primary and secondary) were included in this study. Although the methodology of this study has been published elsewhere [6], we have highlighted the important aspects in this study.

Prior to the commencement of this study, school principals were contacted to explain the aims of this study and its expected outcomes. Upon agreement to take part in this study, school principals completed an informed consent and proceeded to give specific dates for data collection. All eligible students were invited to participate in this study and given a consent form to take home to their parents or guardians and an assent form for assenting to participate in this study. Figure 1 shows the recruitment procedure followed in this study.

### 2.2. Ethical Considerations

The permission to conduct this study was obtained from the Limpopo Provincial Department of Education and the Vhembe District Department of Education, where the schools were located. The Research Ethics clearance for this study was granted by the University of Pretoria Ethics Committee and the Limpopo Provincial Research Ethics Committee under registration (REC 482/2022) and (LPREC/54/2022:PG), respectively. Prior to this study commencing, school principals, parents or guardians, and students signed consent and assent forms, respectively, and were given withdrawal options.

### 2.3. Questionnaire

Data were collected using the English Version of the previously validated ISAAC written questionnaire where school children completed the survey during school hours in the Life Orientations period [18,19]. The medium of instruction used during the data collection at the schools was English. The questionnaire used to collect data contained questions about demographics, respiratory health, exposure to air pollution, and diet on adolescents residing in Vhembe district.

### 2.4. Health Outcomes

The health outcome of asthma was estimated by a positive response to the question “Have you ever had asthma?”.

The health outcome of wheeze was estimated by a positive response to the question “Have you ever had wheezing or whistling in the chest at any time in the past?”.

The health outcome of wheeze in the past was estimated by a positive response to the question “Have you had wheezing or whistling in the chest in the past 12 months?”.

### 2.5. Assessment of Dietary Patterns

In the current study, the ISAAC questionnaire [18,19], which contained 22 food options on everyday dietary intake of children during the last twelve months, was used. It consisted of four food groups, namely starches, fruits and vegetables, junk food, and fats and oils. The frequency of dietary consumption from each food group/category was assessed by answering the following question: “In the past 12 months, how often, on average, did you eat, drink the following? (Please leave blank if you do not know what food it is)”.

There were 3 options as answers, and participants could choose from never or occasionally/once or twice a week/most days of the week for the different foods.

### 2.6. Confounding Variables

The following variables were identified as potential confounders: gender (female/male), community (exposed/non-exposed), born in the study area (yes/no), school (primary/secondary), being a twin (yes/no), type of fuel used in the house for cooking (electricity/gas/paraffin/open fire—wood/coal), type of fuel used in the house for heating (electricity/gas/paraffin/open fire—wood/coal),having pets (dog and/or cat) at home (yes/no), the frequency of trucks passing in the street (never/seldom/frequently during the day/almost whole day), currently smoking tobacco (not at all/less than daily/daily), smoke water pipe (yes/no), vigorous physical activity (never or only occasionally/once or twice per week/most or all days), use of paracetamol (never/at least once a year/at least once a month), and playing social games including social media and/or watching television (less than 1 h/1 h but less than 3 h/3 h but less than 5 h/5 h or more) [8].

### 2.7. Data Management and Statistical Analysis

The data were captured using Research Electronic Data Capture (REDCap) where a template of the questionnaire was created, and responses were captured by qualified data capturers to ensure quality of data. Following data capturing, the data were analyzed using statistical software STATA Version 17 [20]. Descriptive statistics for categorical data were determined by way of frequencies.

Prevalence of asthma; wheeze and wheeze in the past 12 months; frequency of dietary pattern consumption; and confounding variables were calculated by dividing the number of participants who responded “YES” to a particular question by the total number of questionnaires completed. Where observations were not recorded, they were deemed as missing data which consequently caused a difference in the sample size.

The crude and adjusted odds ratios (ORs) and their 95% confidence intervals (CIs) were calculated using the univariate and multivariate logistic regression analysis (MLR) to estimate the degree of association between the frequency of dietary pattern consumption, asthma, wheeze, and wheeze in the past 12 months. For the adjusted OR, to control for confounding variables, variables were put in the initial model, followed by the addition of the potential confounding variable in a stepwise manner. Covariates used in the adjusted model were age, number of older brothers or sisters, and number of years living in the community. In this level of analysis, variables were significant if *p*-values were <0.05.

## 3. Results

### 3.1. Demographic Characteristics of the Study Participants

There were more females (57.89%) than males (42.11%). A majority of the study participants were 13 years old at the time of survey (68.83%) and were born in the area of study (76.06%). Nearly 55% of the participants were in secondary school, and 45.36% were in primary school. A majority of the study participants (62.90%) never or occasionally engaged in physical activity, while a notable number of participants (12.57%) spent five or more hours watching TV daily. Detailed results on the general characteristics of the study participants can be found in a previously conducted study by Rathogwa-Takalani et al. [6].

### 3.2. Health Outcomes and Frequency of Food Consumption Among Study Participants

The prevalence of ever having asthma, ever having wheeze, and wheeze in the past 12 months in this study was 18.91%, 38.06%, and 67.40%, respectively, as shown in Table 1 below. When asked “How often, on average did you eat or drink the following foods?”, a majority (53.88%) of the study participants said they ate bread most or all days. Nearly 50% of the participants said they never ate or had only occasionally eaten seafood. Just over 18% of the participants said they ate different fast food including burgers on most or all days. Approximately half of the participants (53.03%) ate fruits on most of all days of the week.

Several dietary factors were found to be statistically significant risk factors for asthma in the univariate logistic analysis. Consuming seafood on most or all days of the week was a statistically significant risk factor (OR = 1.79; 95% CI 0.65–1.24) for asthma alongside consuming nuts most or all days of the week (OR = 1.06; 95% CI 0.79–1.42). Furthermore, consuming fast foods (OR = 1.32; 95% CI 1.00–1.74) and burgers (OR = 1.41; 95% CI 1.06–1.88) three or more times per week were statistically significant risk factors for asthma as shown with crude odds ratios in Table 2.

In the adjusted model, also in Table 2 below, consumption of seafood most or all days of the week was statistically significantly associated with asthma as a risk factor (OR = 1.60; 95% CI 1.03–2.49). Moreso, consumption of nuts once or twice weekly was also a significant risk factor for asthma (OR = 1.55; 95% CI 1.11–2.17). Lastly, consumption of cereal most or all days was protective of asthma (OR = 0.67; 95% CI 0.45–0.99), although borderline significant (*p* = 0.05).

**Table 2 ijerph-22-00502-t002:** Crude and adjusted odds ratios of asthma and dietary and behavioral patterns among study participants (n = 2855).

Variable	Crude OR		Adjusted OR	
	(95% CI)	*p*	(95% CI)	*p*
**SEAFOOD**				
Never or only occasionally	1		1	
Once or twice per week	1.15 (0.93–1.43)	0.18	1.07 (0.78–1.47)	0.65
Most or all days	1.79 (0.65–1.24)	**<0.001**	1.60 (1.03–2.49)	**0.035**
**FRUIT**				
Never or only occasionally	1		1	
Once or twice per week	0.64 (0.46–0.89)	**0.009**	0.75 (0.41–1.36)	0.34
Most or all days	0.65 (0.48–0.88)	**0.006**	0.81 (0.45–1.45)	0.48
**CEREAL**				
Never or only occasionally	1		1	
Once or twice per week	0.93 (0.72–1.21)	0.62	0.74 (0.50–1.07)	0.11
Most or all days	0.67 (0.51–0.87)	**0.003**	0.67 (0.45–0.99)	**0.050**
**BREAD**				
Never or only occasionally	1		1	
Once or twice per week	0.76 (0.56–1.03)	0.08	0.87 (0.52–1.46)	0.60
Most or all days	0.60 (0.45–1.81)	**0.001**	0.75 (0.45–1.23)	0.26
**PASTA**				
Never or only occasionally	1		1	
Once or twice per week	0.87 (0.69–1.10)	0.26	0.88 (0.62–1.25)	0.48
Most or all days	1.39 (1.06–1.84)	**0.017**	1.46 (0.98–2.19)	0.06
**NUTS**				
Never or only occasionally	1		1	
Once or twice per week	0.85 (0.65–1.12)	**0.027**	1.55 (1.11–2.17)	**0.010**
Most or all days	1.06 (0.79–1.42)	0.67	1.43 (0.94–2.17)	0.08
**FAST FOOD BURGER**				
Never or only occasionally	1		1	
Once or twice per week	1.00 (0.78–1.27)	1.00	1.11 (0.76–1.62)	0.57
Most or all days	1.41 (1.06–1.88)	**0.016**	1.24 (0.78–1.97)	0.34
**FAST FOOD EXCL. BURGER**				
Never or only occasionally	1		1	
Once or twice per week	0.90 (0.70–1.14)	0.39	0.75 (0.51–1.10)	0.15
Most or all days	1.32 (1.00–1.74)	**0.049**	0.88 (0.57–1.36)	0.56
**How many times a week do you engage in vigorous physical activity**				
Never or only occasionally	1		1	
Once or twice per week	3.83 (3.08–4.77)	**<0.001**	2.77 (1.99–3.83)	**<** **0.001**
Most or all days	3.15 (2.33–4.26)	**<0.001**	2.28 (1.42–3.64)	**0.001**
**During a normal week of 7 days, how many hours a day (24 h) do you watch television?**				
Less than 1 h	1		1	
1 h but less than 3 h	1.35 (1.08–1.68)	**0.007**	1.34 (0.94–1.90)	0.10
3 h but less than 5 h	1.10 (0.79–1.53)	0.54	1.42 (0.88–2.29)	0.14
5 h or more	0.93 (0.67–1.30)	0.70	0.90 (0.55–1.49)	0.70
**During a normal week of 7 days, how many hours a day (24 h) do you spend on any social games?**				
Less than 1 h	1		1	
1 h but less than 3 h	1.07 (0.83–1.37)	0.58	0.75 (0.51–1.09)	0.13
3 h but less than 5 h	1.75 (1.35–2.27)	**<0.001**	0.84 (0.55–1.27)	0.41
5 h or more	1.31 (0.97–1.76)	0.07	0.65 (0.41–1.04)	0.07
**Are you twin?**				
No	1		1	
yes	3.62 (2.78–4.70)	**<0.001**	1.83 (1.16–2.85)	**0.008**
**How often do trucks pass through the street where you live on weekdays?**				
Never	1		1	
Seldom	1.46 (1.12–1.90)	**0.005**	1.16 (0.76–1.77)	0.46
Frequently through the day	2.52 (1.94–3.26)	**<0.001**	1.63 (1.04–2.55)	**0.033**
Almost the whole day	1.65 (1.25–2.16)	**<0.001**	1.62 (1.07–2.44)	**0.022**
**In the past 12 months how often, on average, have you taken paracetamol?**				
Never	1		1	
At least once a year	2.73 (2.19–3.40)	**<0.001**	1.07 (0.74–1.53)	0.70
At least once a month	2.50 (1.90–3.29)	**<0.001**	1.91 (1.29–2.81)	**0.001**
**In the past 12 months, have you had a cat in your home?**				
**No**	1		1	
yes	2.22 (0.81–2.71)	**<0.001**	1.22 (0.88–1.69)	0.21
**Do you smoke water pipe?**				
No	1		1	
yes	2.62 (2.01–3.42)	**<0.001**	1.54 (0.95–2.49)	0.07
**What fuel is usually used for cooking?**				
Electricity	1		1	
Gas	1.93 (1.49–2.51)	**<0.001**	0.81 (0.51–1.29)	0.39
Paraffin	3.90 (2.61–5.82)	**<0.001**	0.47 (0.29–0.77)	**0.003**
Open fires	1.19 (0.94–1.51)	0.14	0.65 (0.36–1.19)	0.16
**What fuel is usually used for heating?**				
Electricity	1		1	
Gas	2.09 (1.60–2.72)	**<0.001**	1.18 (0.75–1.84)	0.46
Paraffin	0.93 (0.70–1.24)	0.66	1.69 (0.84–3.38)	0.13
Open fires	0.84 (0.56–1.28)	0.43	1.31 (0.91–1.89)	0.13

Statistically significant *p*-values (<0.05) for the crude OR and the adjusted OR are highlighted in bold. Only risk factors and confounders that showed association with health outcome were included in crude OR and adjusted OR. The model was adjusted for all significant variables from crude OR. Covariates used in the adjusted model were age, number of older brothers or sisters, and number of years living in the community.

Several factors were found to be statistically significant risk factors for ‘wheeze ever’ in the univariate regression analysis as depicted in Table 3 below. Amongst others, consuming olive oil most or all days of the week (OR = 1.40; 95% CI 1.13–1.73), consuming nuts most or all days of the week (OR = 1.88; 95% CI 1.50–2.37) and fast food consumption most or all days of the week (OR = 1.28; 95% CI 1.01–1.62) were significant risk factors of having ever had wheeze.

In the adjusted model, also in Table 3, seafood consumption once or twice per week was protective of ever having wheeze (OR = 0.76; 95% CI 0.59–0.96). Moreso, consumption of fruits once or twice weekly and most or all days of the week was also a protective factor against ever having wheeze (OR = 0.55; 95% CI 0.32–0.94; and OR = 0.59; 95% CI 0.35–1.01). Engaging in vigorous physical activity once or twice weekly (OR = 3.41; 95% CI 2.62–4.45) and most or all days of the week (OR = 3.11; 95% CI 2.07–4.65) was a significant risk factor for ever having wheeze.

Table 4 below shows several statistically significant factors in the univariate regression analysis. Consuming seafood most or all days was a risk factor for wheeze in the past 12 months (OR = 1.61; 95% CI 1.04–2.50). Additionally, consuming butter once or twice a week (OR = 0.66; 95% CI 0.48–0.92) and consuming fruit most or all days (OR = 0.62; 95% CI 0.39–0.98) were also protective factors for wheeze in the past 12 months. In the same model, consuming fast food burgers (OR = 1.54; 95% CI 1.03–2.31) and all other fast foods (OR = 1.67; 95% CI 1.11–2.51) most or all days were significant risk factors for wheeze in the past 12 months.

The adjusted odds ratios show that consuming seafood once or twice weekly was a risk factor for wheeze in the past 12 months (OR = 1.56; 95% CI 1.01–2.40). Consuming other dairy besides milk was also a risk factor for wheeze in the past 12 months (OR = 1.51; 95% CI 1.01–2.25). Protective factors for wheeze in the past 12 months were the following: consuming fruit once or twice weekly (OR = 0.14; 95% CI 0.50–0.37) and most or all days of the week (OR = 0.12; 95% CI 0.04–0.33). Consuming butter either once or twice weekly (OR = 0.55; 95% CI 0.35–0.87) or most or all days of the week (OR = 0.36 95% CI 0.20–0.63) was also a protective factor for wheeze in the past 12 months as shown in Table 4 below.

**Table 4 ijerph-22-00502-t004:** Crude and adjusted odds ratios of wheeze in the past 12 months and dietary and behavioral patterns among study participants (n = 2855).

Variable	Crude OR		Adjusted OR	
	(95% CI)	*p*	(95% CI)	*p*
**SEAFOOD**				
Never or only occasionally	1		1	
Once or twice per week	1.25 (0.94–1.68)	0.27	1.56 (1.01–2.40)	**0.042**
Most or all days	1.61 (1.04–2.50)	**0.030**	1.22 (0.66–2.23)	0.51
**FRUIT**				
Never or only occasionally	1		1	
Once or twice per week	0.76 (0.94–1.23)	0.27	0.14 (0.50–0.37)	**<** **0.001**
Most or all days	0.62 (0.39–0.98)	**0.041**	0.12 (0.04–0.33)	**<** **0.001**
**BUTTER**				
Never or only occasionally	1		1	
Once or twice per week	0.66 (0.48–0.92)	**0.015**	0.55 (0.35–0.87)	**0.012**
Most or all days	0.82 (0.55–1.22)	0.33	0.36 (0.20–0.63)	**<** **0.001**
**OTHER DAIRY**				
Never or only occasionally	1		1	
Once or twice per week	0.94 (0.66–1.32)	0.72	1.35 (0.83–2.22)	0.22
Most or all days	1.51 (1.01–2.25)	**0.040**	2.35 (1.30–4.25)	**0.005**
**FAST FOOD BURGER**				
Never or only occasionally	1		1	
Once or twice per week	1.03 (0.75–1.42)	0.83	0.85 (0.51–1.43)	0.55
Most or all days	1.54 (1.03–2.31)	**0.034**	1.17 (0.61–2.27)	0.62
**FAST FOOD EXCL. BURGER**				
Never or only occasionally	1		1	
Once or twice per week	1.12 (0.81–1.54)	0.46	1.41 (0.85–2.36)	0.17
Most or all days	1.67 (1.11–2.51)	**0.034**	1.21 (0.60–2.06)	0.72
**FIZZY DRINKS**				
Never or only occasionally	1		1	
Once or twice per week	0.56 (0.38–0.83)	**0.004**	0.88 (0.48–1.64)	0.70
Most or all days	0.81 (0.54–1.23)	0.34	1.12 (0.60–2.06)	0.25
**How many times a week do you engage in vigorous physical activity?**				
Never or only occasionally	1		1	
Once or twice per week	2.72 (2.01–3.69)	**<0.001**	2.83 (1.82–4.40)	**<** **0.001**
Three or more times a week	1.67 (1.13–2.46)	**0.010**	1.48 (0.84–2.62)	0.17
**During a normal week of 7 days, how many hours a day (24 h) do you watch television?**				
Less than 1 h	1		1	
1 h but less than 3 h	1.26 (0.93–1.70)	0.12	1.07 (0.67–1.71)	0.76
3 h but less than 5 h	1.64 (1.01–2.66)	**0.042**	1.44 (0.71–2.92)	0.29
5 h or more	1.36 (0.87–2.12)	0.16	0.68 (0.35–1.31)	0.25
**During a normal week of 7 days, how many hours a day (24 h) do you spend on any social games?**				
Less than 1 h	1		1	
1 h but less than 3 h	1.07 (0.77–1.49)	0.64	0.82 (0.49–1.35)	0.44
3 h but less than 5 h	1.90 (1.29–2.78)	**0.001**	0.94 (0.50–1.76)	0.85
5 h or more	1.36 (0.82–1.82)	0.30	0.88 (0.48–1.61)	0.68
**Are you a TWIN?**				
No	1		1	
yes	1.75 (1.17–2.61)	**0.006**	0.93 (0.51–1.70)	0.83
**How often do trucks pass through the street where you live on weekdays?**				
Never	1		1	
Seldom	1.31 (0.92–1.86)	0.12	1.77 (1.03–3.05)	**0.038**
Frequently through the day	2.19 (1.51–3.18)	**<0.001**	2.47 (1.32–4.62)	**0.004**
Almost the whole day	1.28 (0.90–1.83)	0.16	1.43 (0.83–2.48)	0.19
**In the past 12 months, how often, on average, have you taken paracetamol?**				
Never	1		1	
At least once a year	1.69 (1.26–2.28)	**<0.001**	1.06 (0.66–1.71)	0.79
At least once a month	2.29 (1.57–3.62)	**<0.001**	1.29 (0.74–2.27)	0.35
**In the past 12 months, have you had a cat in your home?**				
No	1		1	
yes	0.43 (1.69–3.07)	**<0.001**	1.66 (1.05–2.63)	**0.029**
**What fuel is usually used for cooking?**				
Electricity	1		1	
Gas	1.27 (0.88–1.83)	0.18	0.93 (0.51–1.72)	0.84
Paraffin	2.88 (1.42–5.82)	**0.003**	1.53 (0.44–5.29)	0.49
Open fires	1.71 (1.22–2.40)	**0.002**	2.60 (1.57–4.32)	**<0.001**

Statistically significant *p*-values (<0.05) for the crude OR and the adjusted OR are highlighted in bold. Only risk factors and confounders that showed an association with the health outcome were included in the crude OR and adjusted OR. The model was adjusted for all significant variables from the crude OR. Covariates used in the adjusted model were age, number of older brothers or sisters, and number of years living in the community.

## 4. Discussion

This study investigated the association between the frequency of dietary pattern consumption and environmental factors and asthma and wheeze among adolescents in Vhembe district, Limpopo Province, South Africa. The prevalence of asthma, ever having wheeze, and wheeze in the past 12 months were 18.91%, 38.06%, and 67.40%, respectively. Asthma prevalence in this study was higher than that of a similar study conducted among adolescents by Nkosi et al. [8] and Mphahlele et al. [21] in the North West Gauteng and KwaZulu Natal Provinces. It is worth noting that the prevalence of wheeze (38.2%) in the Nkosi et al. [8] study was nearly equal to the wheeze prevalence in the current study [8] but still higher than that of the Mphahlele and colleagues [21] study. The higher prevalence in the current study may be because Vhembe District is a malaria endemic area that is exposed to malaria eradication programs such as Indoor Residual Spraying. Studies have shown concerns regarding the safety and possible adverse health effects of exposure to the complex chemical mixtures used in IRS [22,23]. The marked increase in the prevalence of asthma is alarming and is fast becoming a major public health concern.

The literature has highlighted the importance of diet in the increase in asthma and respiratory symptoms such as wheeze [24,25,26]. The world is evolving fast, and we are observing a shift in dietary patterns where developing countries have increased the consumption of a westernized diet [8]. In this study, 41.44% of participants consumed fast food once or twice weekly, and nearly 20% affirmed that they consumed fast food most or all days of the week. Moreover, this consumption of fast food was positively associated with the likelihood of presenting with asthma, ever having wheeze, and wheeze in the past.

In a previously conducted study, fast food was consumed more than once a week by 72% of participants [8]. The source of major discrepancies could be that the current study was conducted in a rural setting where access to fast food may be limited. Nevertheless, an unhealthy diet plays an important role in respiratory health, and healthier diets should be encouraged. A study conducted found that eating fast food most or all days of the week was significantly associated with asthma and wheeze as a risk factor in adolescents in South Africa [8].

In this study, just over half of the participants (55.95%) consumed fruit most days of the week, and less than half (39.78%) consumed cereal most days of the week. This response may be attributed to the school feeding scheme that provides different fruits and cereal to learners every day to reduce hunger and encourage a well-balanced diet [27]. In the current study, consuming fruits was negatively associated with the likelihood of presenting with asthma. Participants who consumed fruits had a 45% lower likelihood of ever having had wheeze and a nearly 90% lower likelihood of wheeze in the past 12 months. A study by Nagel and colleagues [28] found that frequent consumption of fruits was associated with a lower lifetime prevalence of asthma [28]. Moreover, a randomized clinical trial showed that consumption of fruits did not affect the exacerbation of asthma or symptoms; however, it showed improvements in the lung function of study participants [29]. The literature shows that fruit consumption is contrary to wheeze prevalence and asthma severity [29,30,31]. Fruits are major components of the Mediterranean diet and have antioxidant properties that have been shown to exert a protective effect on diseases including asthma [15]. Hence, a high-fruit diet may be an additional strategy for the management of asthma and asthma symptoms.

In the current study, seafood was an associated factor across all health outcomes (asthma, ever having wheeze, and wheeze in the past 12 months). Seafood consumption was associated with a lower odds ratio for wheeze and a higher odds ratio for asthma and wheeze in the past 12 months. While seafood is beneficial due to its omega-3 fatty acids, research has shown potential molecular mechanisms in which seafood contributes to asthma pathophysiology. Seafood allergies caused by different fish species are common in adolescents [32]. The allergic response occurs when certain proteins, i.e., parvalbumins, tropomyosin, and other allergens, are recognized by IgE antibodies after exposure [33,34]. In response to this exposure, antigen-specific antibodies are produced that stimulate an allergic response in the form of airway hyperresponsiveness, airway inflammation, or mucus hypersecretion [33,34,35,36].

Contrary to our asthma findings, other studies have demonstrated that daily intake of fish had protective effects against asthma [15,16,17,18,19,20,21,22,23,24,25,26,27,28,29,30,31,32,33,34,35,36,37]. Moreover, a study conducted by Calder et al. classified fish and seafood as rich sources of omega-3 fatty acids that exert anti-inflammatory properties and decrease the risk of asthma and allergic diseases [38]. Furthermore, our findings are in contrast with results from Nagel et al., which showed that seafood was associated with a lower lifetime prevalence of asthma [28]. Whereas a study conducted by Calder et al. classified fish and seafood as rich sources of omega-3 fatty acids that exert anti-inflammatory properties and decrease the risk of asthma and allergic diseases [38]. Epidemiological studies conducted on seafood and its association with asthma and wheeze are inconsistent; although, we can note that a majority of the studies show protective effects [38,39,40].

The use of oils, that is, olive oil and butter, were associated with wheeze in this study. In this study, consuming olive oil was associated with an increased likelihood of wheeze, whereas consuming butter was associated with lower odds of wheeze in the past. Contrary to this, in most studies conducted, healthy fatty acids appear to have protective effects on asthma disease and symptoms [1]. A large-scale cross-sectional study on children assessing margarine butter and olive oil consumption showed that butter consumption once or twice weekly may be inversely associated with asthma and its symptoms in children [41]. Another study also indicated that good fatty acids may positively affect respiratory health [42]. These effects are attributed to the anti-inflammatory properties of certain fatty acids in healthy oils which modulate the body’s inflammatory response [43]. Olive oil in this study was negatively associated with wheeze [1]. This finding is consistent with findings in other studies where olive oil is proven to exert anti-inflammatory effects in the body leading to a protective effect against respiratory diseases, in this case asthma and asthma symptoms [43]. Butter contains saturated fats and polyunsaturated fatty acid-6 (PUFA-6) similar to those used in the preparation of fast foods that are known to have pro-inflammatory effects [1,44]. Asthma pathophysiology involves a T helper cell type 2 (Th-2) response and the production of specific Immunoglobin E (IgE) which leads to airway hyperresponsiveness and allergic sensitization [45]. PUFA-6 results in the pro-inflammatory activity that promotes a type 2 helper T (Th-2) immune response [1,46].

Consumption of nuts has been deemed a source of healthy fats, fiber, and good nutrition. However, some studies have associated it with exacerbation of asthma and its symptoms [14]. In respiratory health, nuts are said to be allergens that may induce an allergic reaction [1,47,48]. This concurs with the findings of the current study, where consumption of nuts was associated with an increased likelihood of asthma and wheeze. In a cohort study of children with allergies, nut allergy was found to have significantly greater odds in relation to having asthma [49]. Although nuts are regarded as nutritious, studies have shown adverse effects associated with nut allergies, and prevention of those effects are of the upmost importance [50].

Some studies have shown that the consumption of dairy products, in particular cow milk, increased positive effects related to asthma and respiratory symptom outcomes [51,52]. Contrary to this, some studies have shown that consumption of dairy products resulted in negative asthma-related outcomes [53,54]. In this study, participants who consumed dairy products most or all days of the week had two times higher odds of wheeze in the past 12 months. The mechanisms by which dairy products have effects on the development of asthma and respiratory symptoms are unclear. However, research has mentioned an inflammatory pathway that is activated when consuming dairy and that plays a role in asthma development [1,54]. Mediators such as Th-1 and Th-17 are involved in neutrophilic asthma, and the main cytokines reported in this pathway are IL-6, IL8, and IL17 which are produced by the Th-17 response [1,55]. Controlling the intake of dairy products may be considered in the management strategies for asthma and asthma-related symptoms.

In this study, engaging in physical activity was associated with an increased likelihood of asthma, wheeze, and wheeze in the past 12 months. Asthma is known to be a barrier of exercise; when symptoms worsen and asthma is not controlled, there is a reduced possibility that asthmatics will participate in physical activity, and this may be due to Exercise-Induced Bronchoconstriction (EIB), a sign of uncontrolled asthma [56]. Other significant factors reported in this study were the use of paracetamol, which was associated with an increased risk of asthma and wheeze, using sources for fuel such as gas and paraffin, which were associated with lower odds, and using open fires which was associated with two times higher odds of asthma and wheeze. Smoking a tobacco water pipe was associated with higher odds of ever having wheeze. Although this study did not focus on air pollution sources in relation to asthma and wheeze, it is important to highlight the significance of air pollution, particularly from smoking tobacco and/or a water pipe. Tobacco smoke exposure from active smoking, a water pipe, or second-hand smoke has been well documented as an important contributor to respiratory conditions, including asthma and wheeze [57,58,59]. Smoking cessation may be considered in the prevention and control of asthma and wheeze [58].

In this study, using gas as a fuel for cooking was associated with a decreased probability of wheeze. These findings were unexpected due to several studies that have shown that fumes from gases were a risk factor for asthma and wheeze [60,61,62]. An asthma report also showed that when used as a cooking fuel, gas releases chemicals that induce inflammation of the airways and may lead to asthma exacerbation [63].

Inconsistent findings observed between the current study and other studies conducted may be due to the difference in the study setting and population, study design, questionnaire used, or exposure variables considered.

## 5. Strengths and Limitations

The current study has some limitations. Due to the nature of a cross-sectional study design, we are unable to prove causation. Furthermore, the use of self-reported dietary data may introduce bias [64]. Therefore, future clinical research is needed to consolidate the association between asthma, asthma symptoms, and diet to better understand the effect that different dietary patterns may have. Lastly, the measures used in this study may not have been as significant or sensitive as desired; however, they have provided baseline data. Hence, another study using the validated FFQ that is designed to cover specific items and information on the different food items may be required to confirm the associations in the current study. This study used multiple logistic regression analysis which is powerful because it allows for the control of several factors simultaneously and provides a better understanding of how each predictor variable affects the outcome.

## 6. Conclusions

This study investigated the association between the consumption of different foods and environmental factors and asthma and wheeze. Our study found that the consumption of cereal, seafood, and fruits had protective effects against asthma and wheeze symptoms. On the contrary, the consumption of nuts was associated with an increased risk of asthma and wheeze symptoms. Future epidemiological studies are required to better understand the causal relationship that may exist between different foods and asthma and wheeze. Furthermore, based on the current study findings, dietary modifications may play a vital role in mitigating asthma and wheeze symptoms and improving respiratory health.

## Figures and Tables

**Figure 1 ijerph-22-00502-f001:**
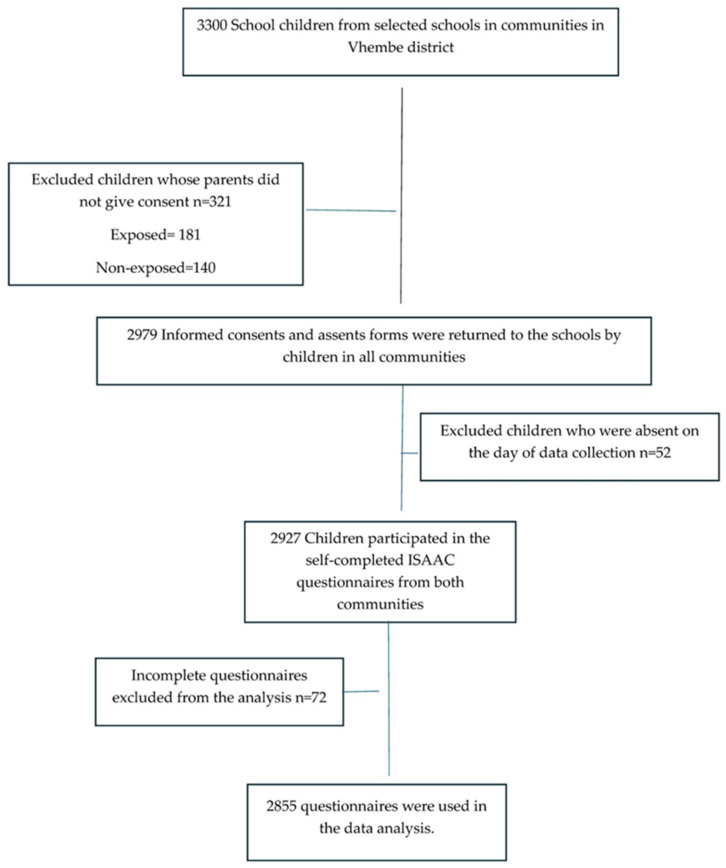
Recruitment procedure followed in this study [6].

**Table 1 ijerph-22-00502-t001:** Prevalence of asthma, ever having wheeze, and wheeze in the past 12 months and the frequency of dietary factors among study participants (n = 2855).

Variable	N	Percentage (%)
**Asthma Ever**		
Yes	533	18.67 (18.91)
No	2286	80.07 (81.09)
**Wheeze Ever**		
Yes	1076	37.69 (38.06)
No	1751	61.33 (61.94)
**Wheeze in the Past 12 Months**		
Yes	705	24.69 (67.40)
No	341	11.94 (32.50)
**Bread**		
Never or only occasionally	357	12.50 (13.52)
Once or twice per week	861	30.16 (32.60)
Most or all days	1423	49.84 (53.88)
**Meat**		
Never or only occasionally	320	11.21 (11.89)
Once or twice per week	1401	49.07 (52.04)
Most or all days	971	34.01 (36.07)
**Seafood**		
Never or only occasionally	1185	41.51 (45.19)
Once or twice per week	1129	39.54 (43.06)
Most or all days	308	10.79 (11.75)
**Raw vegetables**		
Never or only occasionally	931	32.61 (36.27)
Once or twice per week	1089	38.14 (42.42)
Most or all days	547	19.16 (21.31)
**Butter**		
Never or only occasionally	805	28.20 (31.31)
Once or twice per week	1178	41.26 (45.87)
Most or all days	588	20.49 (22.78)
**Olive oil**		
Never or only occasionally	980	34.33 (39.72)
Once or twice per week	918	32.15 (37.21)
Most or all days	569	19.93 (23.06)
**Fast food burgers**		
Never or only occasionally	767	26.87 (29.82)
Once or twice per week	1304	41.44 (50.70)
Most or all days	501	18.04 (19.48)
**Fast food excl. burgers**		
Never or only occasionally	786	27.53 (31.64)
Once or twice per week	1183	41.44 (47.62)
Most or all days	515	18.04 (20.73)
**Fizzy drinks**		
Never or only occasionally	492	17.23 (19.00)
Once or twice per week	1139	39.89 (43.99)
Most or all days	958	33.56 (37.00)
**Other dairy**		
Never or only occasionally	648	22.70 (25.76)
Once or twice per week	1238	43.36 (49.21)
Most or all days	630	22.07 (25.04)
**Fruit**		
Never or only occasionally	270	9.46 (9.98)
Once or twice per week	922	32.29 (34.07)
Most or all days	1514	53.03 (55.95)
**Potatoes**		
Never or only occasionally	558	19.54 (21.60)
Once or twice per week	1357	47.53 (52.54)
Most or all days	668	23.40 (25.86)
**Nuts**		
Never or only occasionally	901	31.56 (35.29)
Once or twice per week	1180	41.33 (46.22)
Most or all days	472	16.53 (18.49)
**Cereal**		
Never or only occasionally	559	19.58 (22.35)
Once or twice per week	947	33.17 (37.86)
Most or all days	995	34.85 (39.78)
**Pasta**		
Never or only occasionally	859	30.09 (33.83)
Once or twice per week	1196	41.89 (47.11)
Most or all days	484	16.95 (19.06)

**Table 3 ijerph-22-00502-t003:** Crude and adjusted odds ratios of ever having wheeze and dietary and behavioral patterns among study participants (n = 2855).

Variable	Crude OR		Adjusted OR	
	(95% CI)	*p*	(95% CI)	*p*
**CEREAL**				
Never or only occasionally	1		**1**	
Once or twice per week	0.88 (0.71–1.09)	0.24	0.80 (0.58–1.09)	0.17
Most or all days	0.80 (0.64–0.99)	**0.042**	0.75 (0.54–1.05)	0.09
**BREAD**				
Never or only occasionally	1		**1**	
Once or twice per week	0.75 (0.58–0.96)	**0.026**	0.74 (0.47–1.15)	0.19
Most or all days	0.79 (0.63–1.01)	0.06	0.95 (0.61–1.47)	0.84
**PASTA**				
Never or only occasionally	1		**1**	
Once or twice per week	1.05 (0.88–1.27)	0.56	1.18 (0.90–1.56)	0.21
Most or all days	1.30 (1.03–1.64)	**0.022**	0.91 (0.63–1.30)	0.61
**RICE**				
Never or only occasionally	1		**1**	
Once or twice per week	0.65 (0.52–0.81)	**<0.001**	0.97 (0.67–1.41)	0.89
Most or all days	0.98 (0.77–1.26)	0.83	1.02 (0.67–1.54)	0.90
**OLIVE OIL**				
Never or only occasionally	1		**1**	
Once or twice per week	1.06 (0.88–1.28)	0.50	1.17 (0.89–1.53)	0.23
Most or all days	1.40 (1.13–1.73)	**0.002**	1.48 (1.09–2.00)	**0.027**
**OTHER DAIRY**				
Never or only occasionally	1		**1**	
Once or twice per week	1.05 (0.86–1.28)	0.59	1.17 (0.87–1.59)	0.28
Most or all days	1.28 (1.02–1.61)	**0.028**	1.21 (0.85–1.73)	0.27
**NUTS**				
Never or only occasionally	1		1	
Once or twice per week	1.21 (1.00–1.45)	**0.046**	1.14 (0.87–1.49)	0.32
Most or all days	1.88 (1.50–2.37)	**<0.001**	1.88 (1.33–2.66)	**<** **0.001**
**FAST FOOD**				
Never or only occasionally	1		1	
Once or twice per week	1.01 (0.84–1.22)	0.88	0.97 (0.73–1.29)	0.84
Most or all days	1.28 (1.01–1.62)	**0.037**	0.91 (0.63–1.31)	0.62
**MEAT**				
Never or only occasionally	1		1	
Once or twice per week	0.77 (0.60–0.99)	**0.050**	0.90 (0.58–1.41)	0.66
Most or all days	1.01 (0.77–1.30)	0.95	0.88 (0.55–1.40)	0.59
**SEAFOOD**				
Never or only occasionally	1		1	
Once or twice per week	0.99 (0.84–1.17)	0.93	0.76 (0.59–0.96)	**0.027**
Most or all days	1.33 (1.03–1.72)	**0.028**	0.88 (0.59–1.30)	0.52
**FRUIT**				
Never or only occasionally	1		1	
Once or twice per week	0.66 (0.50–0.87)	**0.004**	0.55 (0.32–0.94)	**0.031**
Most or all days	0.78 (0.59–1.01)	0.06	0.59 (0.35–1.01)	0.06
**VEGERAW**				
Never or only occasionally	1		1	
Once or twice per week	1.17 (0.97–1.40)	0.09	1.15 (0.88–1.49)	0.29
Most or all days	1.32 (1.06–1.64)	**0.013**	1.26 (0.90–1.74)	0.16
**Fuel used for cooking in the house**				
Electricity	1		1	
Gas	1.44 (1.15–1.81)	**0.001**	0.67 (0.47–0.96)	**0.033**
Paraffin	2.00 (1.36–2.93)	**<0.001**	0.67 (0.35–1.27)	0.22
Open fires (wood, coal)	1.03 (0.85–1.24)	0.75	0.84 (0.63–1.10)	0.21
**Frequency of trucks passing in the street**				
Never	1		1	
Seldom (not often)	1.22 (0.99–1.50)	**0.051**	0.71 (0.51–0.99)	**0.046**
Frequently through the day	1.90 (1.53–2.37)	**<0.001**	0.94 (0.65–1.35)	0.74
Almost the whole day	1.14 (0.87–1.49)	**<0.001**	0.88 (0.63–1.22)	0.44
**Smoke water pipe**				
No	1		1	
Yes	3.00(2.33–3.86)	**<0.001**	1.70 (1.10–2.63)	**0.016**
**Have a cat at home**				
No	1		1	
Yes	1.84 (1.55–2.18)	**<0.001**	1.04 (0.79–1.36)	0.77
**Use paracetamol**				
Never	1		1	
At least once a year	2.63(2.18–3.16)	**<0.001**	1.73 (1.30–2.30)	**<** **0.001**
At least once a month	2.00 (1.58–2.54)	**<0.001**	1.84 (1.32–2.57)	**<** **0.001**
**Physical activity**				
Never or only occasionally	1		1	
Once or twice per week	4.31 (3.57–5.20)	**<0.001**	3.41 (2.62–4.45)	**<0.001**
Most or all days	4.12 (3.17–5.37)	**<0.001**	3.11 (2.07–4.65)	**<** **0.001**
**Games social**				
Less than 1 h	1		1	
1 h but less than 3 h	1.17 (0.97–1.43)	0.09	1.18 (0.87–1.59)	0.27
3 h but less than 5 h	1.43 (1.15–1.78)	**0.001**	1.09 (0.76–1.54)	0.62
5 h or more	1.31 (1.03–1.66)	**0.024**	1.33 (0.92–1.92)	0.12
**Watching Television**				
Less than 1 h	1		1	
1 h but less than 3 h	1.21 (1.01–1.45)	**0.032**	1.18 (0.89–1.57)	0.22
3 h but less than 5 h	0.90 (0.69–1.17)	0.45	1.16 (0.78–1.72)	0.44
5 h or more	0.99 (0.77–1.28)	0.96	1.04(0.71–1.52)	0.82

Statistically significant *p*-values (<0.05) for the crude OR and the adjusted OR are highlighted in bold. Only risk factors and confounders that showed association with health outcome were included in crude OR and adjusted OR. The model was adjusted for all significant variables from crude OR. Covariates used in the adjusted model were age, number of older brothers or sisters, and number of years living in the community.

## Data Availability

The ethical approval we received limits us from sharing the data publicly. The raw data analyzed are available upon reasonable request to the authors.

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
