# Peer review of "Dietary Patterns in Relation to Asthma and Wheeze Among Adolescents in a South African Rural Community"

_ijerph, 2025, doi:10.3390/ijerph22040502_

Round 1
Reviewer 1 Report
Comments and Suggestions for Authors
To the Authors
This cross-sectional study investigated dietary and environmental factors associated with asthma/or wheezing in African adolescents.
Overall, a good effort was made by the authors. However, there are a number of issues that require clarification and revision. As in its present state, this article does not represent a scientifically sound study.
Please refer to my comments to the authors.
Comments
ABSTRACT
Given that associations between environmental factors and asthma/wheeze were also assessed in this study, add this to the aim of the abstract and main manuscript.
MAIN MANUSCRIPT
METHODS
-Line 110 Please change the word ‘learners’ to students
-Line 115 ‘Data were collected using the English Version of the previously validated ISAAC written questionnaire where school children completed the survey during school hours in the Life Orientations period.’
-Lines 127-129 ‘In the current study, the ISAAC questionnaire which contained 22 food options on everyday dietary intake of children during the last twelve months was used. It consisted of four food groups, namely: starches, fruits and vegetables, junk food, fats, and oils.’
Please add a reference to where the ISAAC questionnaires (asthma and food questionnaires) can be found and add them in an appendix or as an online supplement.
2.6 Confounding variables
-Add references to support why these variables were chosen as confounding factors.
Statistical Analysis
-Line 150 ‘Following data capturing, the data was analyzed using statistical software STATA Version 17.’ Add ref for STATA software
-Line 159 ‘Univariate’, typo error. Lower case
-In the statistical analysis, mention the covariates that were used in the adjusted regression models.
-Lines 163-165, ‘Multiple logistic regression analysis is powerful because it allows you to control several factors simultaneously and provides a better understanding of how each predictor variable affects the outcome.’ Add this statement to ‘Strengths/limitations’ of the Discussion.
RESULTS
-Line 182 ‘Majority of the participants (53.03%)………’
Please revise. 53% of the sample does not represent the ‘majority’ but approximately ‘half’.
Table 1
-In order to shorten Table 1, it is sufficient to show the row ‘yes’ for ‘asthma ever’, ‘wheeze’, ‘wheeze in the past 12 months’. Delete the row for ‘missing’
Tables 2-4
-The legend should describe the data being presented. For example, in table 2 dietary and behavioral patterns of the study population.
-Please revise the legends of Tables 2-4
-Beneath tables, note that statistically significant P-values are indicated in bold text and which factors were added to the adjusted regression models
-in Table 4 row ‘Yes cat in your home’, p=0.029 is statistically significant and should be in ‘bold text’.
DISCUSSION
The point of the discussion is to summarize the results of the multivariate analysis, providing feasible explanations for the observations supported by evidence from the literature. Please revise.
Line 236 ‘The study investigated the association between the frequency of dietary pattern consumption with asthma and wheeze among adolescents in Vhembe district…….
Be more precise. According to data in Tables, dietary as well as environmental factors associated with asthma and wheeze were investigated. Revise line 236
-lines 238-243 ‘The prevalence of asthma, wheeze ever and wheeze in the past 12
months were 18.91%, 38.06% and 67.40% respectively. Asthma prevalence in this study
was higher than that of a similar study conducted among adolescents by Nkosi et al. [5] and Mphahlele et al. [15] in North-West, Gauteng ………the wheeze prevalence in the current study [5], but still higher than that of Mphahlele colleagues [15] study’.
Give a feasible reason why the prevalence of asthma and wheeze was higher in the present study, e.g. industrial region.
-Line 261, 54% consuming fruit and 40% cereal, are not considered to be high consumption rates. Only half of the sample consumed fruit and less than half cereal. Revise.
-Line 272, ‘Fruits are major components of the Mediterranean diet …..
In Africa, I believe that the Mediterranean diet is not a dietary pattern adhered by this particular population. I think that a healthy, balanced diet would be more appropriate. Please correct me if I am wrong, I am not an expert on African culinary habits.
Please change the reference accordingly.
-Line 276 ‘In the current study, seafood was an associated factor across all health outcomes (asthma, wheeze ever and wheeze in the past 12 months)’. Provide an explanation why seafood might be positively associated with asthma and wheeze in your study. Possible molecular mechanisms in association with asthma pathophysiology. Similarly, for olive oil and butter (line 292).
-Line 298 Another study also indicated that good fatty acids may positively affect respiratory health [30]. Reference 30, the article by Wypych et al. discusses the pro-inflammatory effects of saturated fats and anti-inflammatory of omega3 fatty acids etc. in relation to respiratory disease and microbiota. Given that microbiota was not assessed in your study, I believe that this reference does not support your findings.
Regarding the results of your study, from a molecular perspective, elaborate how butter (a source of saturated fats and PUFA- 6 fatty acids) as well as olive oil (rich in monounsaturated fats) might be associated with asthma and wheeze. Provide a suitable study in support of your explanation.
-Line 322 ‘However, research has mentioned an inflammatory pathway present when consuming dairy, hat plays a role in asthma development’
Elaborate, mention the mediators involved in the pro-inflammatory pathway associated with asthma induction i.e., Th17, IL-17 etc.
-Line 326 ’In this study, physical activity was a significant risk factor associated with asthma, wheeze, and wheeze in the past. Asthma is known to be a barrier of exercise, when symptoms worsen and asthma is not controlled, there is a less possibility that asthmatics will participate in physical activity (41). ‘
Ref 41 Lawson et al. found that overweight children with asthma were less likely to participate in exercise. From your study, it appears that regular exercise was assessed in children regardless of weight status. I suggest that a more appropriate reference is added to validate your findings.
One other explanation, is exercise-induced asthma which is common in pediatric patients with asthma.
-Line 329 ‘Other significant factors of asthma and wheeze reported in this study were the use of paracetamol and air pollution sources such as using gas, paraffin and open fires as fuel, smoking tobacco and water pipe’
According to data in Table 3, from the adjusted analysis, ‘using gas as cooking fuel’ was negatively associated with wheeze ever [OR: 0.67(0.47-0.96) P= 0.033]. Mention and explain this in the discussion.
-Line 340, ‘A review conducted by Lv et al. showed that studies conducted in pregnant women and adults failed to show associations between dietary patterns and asthma.’
In the context that your study focuses on childhood asthma, then recent studies undertaken in children and adolescents should be used to back up your findings and not those referring to infancy or adulthood. Delete line 340 and revise line 342.
STRENGTHS AND LIMITATIONS
-Line 346 ‘Furthermore, the use of self-reported dietary data may introduce bias.’
Add a reference to support this statement.
CONCLUSION
-Line 355 Environmental factors were also assessed, add this.
-Line 356 ‘Our study found that certain foods had protective effects against asthma and wheeze symptoms.’ Be specific, list the foods that have a protective effect on asthma/wheeze.
Reviewer 2 Report
Comments and Suggestions for Authors
This document contains a study on the relationship between asthma and wheeze in a South African community. The study shows how the type of diet has an effect on both asthma and whezze. The study requires some changes that could contribute to its improvement.
1. Please, each citation need to be carefully reviewed, for example for the definition of asthma and epidemiology, citations about reviews on asthma and diet are used. It is likely that it is not the direct source of the definition or epidemiology. In fact, reviewing the rest of the document, this happens on several occasions, since articles are cited that have no relation to the text.
2. What do you mean by “Expose” and “Nonexposed” in figure 1?
3. Because the study is based on questionnaires, please indicate the reference where the validation is shown.
4. Check the use of capital letters, for example in section 3.2 of Results, Asthma and Wheeze appear in capital letters, but it is not necessary. This is observed in several parts of the article.
Round 2
Reviewer 1 Report
Comments and Suggestions for Authors
TO THE AUTHORS
I thank the authors for the meticulous revision. However, there are a number of issues that require clarification in order to improve the study comprehensibility and transparency.
As in its current state, this manuscript does not represent a scientifically-sound study.
Please refer to my comments to the authors.
METHODS
-Line 218’ Multiple logistic regression analysis is powerful because it allows you to control several factors simultaneously and provides a better understanding of how each predictor variable affects the outcome.’ Move this statement to the strengths/limitations and delete it from line 218.
RESULTS
In general, regarding the presentation of p-values in tables and text.
Statistically significant p-values should be reported to 3-decimal places e.g P= 0.024, and those non-significant to 2-decimals e.g P= 0.456 as P= 0.46.
Revise throughout the manuscript as well as in tables.
SECTION 3.2,
-Line 234, ‘majority of study participants said they ate bread most or all days.
According to Table 1, 53.88% of participants ate bread ‘most or all days’.
In text, next to ‘majority’ include in brackets (53.88%) as follows:
‘Majority (53.88%) of study participants said they ate bread most or all days’.
-Line 298 ‘Lastly, consumption of cereal most or all days was protective of asthma (OR= 0.67 95% CI 0.45- 0.99).’
According to Table 2, the p-value for the consumption of cereal ‘on most or all days’ was P = 0.05, that is borderline. Include this in text as follows:
-Line 298 ‘Lastly, consumption of cereal most or all days was protective of asthma (OR= 0.67 95% CI 0.45- 0.99), although borderline significant (P=0.05).’
TABLES 2-4
Beneath Tables 2-4, list the covariates used in the adjusted regression analysis as follows:
b Regression model adjusted for age, number of siblings and number of years living in the community.
DISCUSSION
-Line 401, statistical data (ORs, 95%CI) should not be repeated in the discussion section.
However, the effect size of the observed associations can be described in text as lower, reduced, increased, positive or negative associations in terms of the outcome.
For example, on line 401, from the adjusted regression analysis consumption of fruits vs wheeze ever (OR: 0.55) can be described as 45% lower odds/or likelihood of wheeze ever or a ‘negative’ association on ‘wheeze ever’.
Delete all ORs, 95%CI from the discussion section and revise: lines 436, 437, 457, 504, 514, 523, 524, 528-531, 540.
-Line 451, ‘lower lifeline of asthma’. Lifeline? Do you mean lifetime of asthma?
If yes, please revise.
-Line 503. ‘This concurs with the findings of the current study, where consumption of nuts was negatively associated with asthma (OR= 1.55 95% CI 1.11-2.17) 504 and wheeze (OR=1.88 95% CI 1.33-2.66).’
This is incorrect. According to your findings, nut intake was associated with increased likelihood of asthma and wheeze, i.e positively associated with asthma and wheeze.
Negatively associated implies that as the independent variable increases the dependent outcome decreases or is associated with lower odds, which is not the case with nut consumption.
Revise accordingly throughout the discussion.
-Line 514, OR: 2.35 can be described as more than double or about two times higher odds of wheeze in the past 12 months
The same for lines 523, 524, 528, 530, 531.Delete all ORs and 95%CI and interpret as suggested above on line 514.
REFERENCES
Names of journals should be abbreviated as they appear in PUBMED.
For example, Pediatric Allergy and Immunology as Ped Allergy Immunol.
Round 3
Reviewer 1 Report
Comments and Suggestions for Authors
To the Authors
There are few errors that require attention before this manuscript represents a scientifically sound study and is in accordance with the journal format.
See this as an opportunity to master the art of writing for scientific journals.
We look forward to future submissions from this research group.
DISCUSSION
Lines 278, 279 Remove the ORs: 0.55, 0.12 from the text
Line 352, typo error ‘wheeze in the past 12 months’ not ‘wheeze in the past’ ?
LINE 367, Remove OR: 0.67 from the text
REFERENCES
Ref 13 Abbreviate the journal name as Pediatr Allergy Immunol
Ref 28. Please check this reference.
Perhaps it is the same study that was published in Thorax 2010?.
Nagel G, Weinmayr G, Kleiner A, Garcia-Marcos L, Strachan DP; ISAAC Phase Two Study Group. Effect of diet on asthma and allergic sensitisation in the International Study on Allergies and Asthma in Childhood (ISAAC) Phase Two. Thorax. 2010 Jun;65(6):516-22. doi: 10.1136/thx.2009.128256.
Ref 31 Mendez et al. Pediatr Allergy Immunol and NOT PAI
Re 38 Calder et al. Proc Nutr Soc NOT PNS
Ref 39 Revise the journal name as Curr Allergy Asthma Rep
Ref 44. Cardinale F, Tesse R, Fucilli C, Loffredo MS, Iacoviello G, Chinellato I, Armenio L. Correlation between exhaled 531 nitric oxide and dietary consumption of fats and antioxidants in children with asthma.
Please add Journal details
Ref 51 Loss et al, J Allergy Clin Immunol. NOT JACI
Revise accordingly
Ref 50 Han et al, J Allergy Clin Immunol. NOT JACI
